# Losing the Arms Race: Greater Wax Moths Sense but Ignore Bee Alarm Pheromones

**DOI:** 10.3390/insects10030081

**Published:** 2019-03-23

**Authors:** Yuan Li, Xingchuan Jiang, Zhengwei Wang, Junjun Zhang, Katrina Klett, Shahid Mehmood, Yufeng Qu, Ken Tan

**Affiliations:** 1CAS Key Laboratory of Tropical Forest Ecology, Xishuangbanna Tropical Botanical Garden, Chinese Academy of Sciences, Kunming 650000, China; liyuan@xtbg.ac.cn (Y.L.); zhangjunjun@xtbg.ac.cn (J.Z.); Katrina.klett@gmail.com (K.K.); shahid@xtbg.ac.cn (S.M.); qu_yu_feng@163.com (Y.Q.); 2College of Life Sciences, University of Chinese Academy of Sciences, Beijing 100049, China; 3College of Plant Protection, Anhui Agricultural University, Hefei 230036, China; jxc678@sina.cn

**Keywords:** electroantennogram, *Galleria mellonella*, oviposition site preference, suitable habitat hypothesis, alarm pheromones

## Abstract

The greater wax moth, *Galleria mellonella* L., is one of main pests of honeybees. The larvae burrow into the wax, damaging the bee comb and degenerating bee products, but also causes severe effects like driving the whole colony to abscond. In the present study, we used electroantennograms, a Y maze, and an oviposition site choice bioassay to test whether the greater wax moth can eavesdrop on bee alarm pheromones (isopentyl acetate, benzyl acetate, octyl acetate, and 2-heptanone), to target the bee colony, or if the bee alarm pheromones would affect their preference of an oviposition site. The results revealed that the greater wax moth showed a strong electroantennogram response to these four compounds of bee alarm pheromones even in a low concentration (100 ng/μL), while they showed the highest response to octyl acetate compared to the other three main bee alarm components (isopentyl acetate, benzyl acetate, and 2-heptanone). However, the greater wax moth behavioral results showed no significant preference or avoidance to these four bee alarm pheromones. These results indicate that bees are currently losing the arms race since the greater wax moth can sense bee alarm pheromones, however, these alarm pheromones are ignored by the greater wax moth.

## 1. Introduction

The greater wax moth, *Galleria mellonella* (GWM), is a natural pest of honey bee colonies, which has already been reported to be harmful to *Apis mellifera* [1], *Apis cerana* [2] and *Apis dorsata* [3]. Its larvae burrow into the comb of a honeybee hive, which causes damage to the comb or even causes *A. cerana* colony absconding [4,5]. GWMs lay eggs in stored combs which may hatch later when the temperature warms [6,7,8]. That is why the GWM has been recognized as one of the main pests to stored combs.

The developmental durations for GWM eggs, larvae and pupae have been reported to be around five days, six to seven weeks, and two weeks respectively at 29–32 °C [1]. As the larvae develop into pupae they progress through eight stages [5]. The larval development process can be prolonged up to 12 weeks at room temperature which is around 24 °C. The adult’s life span is about two to five weeks depending on temperature and other conditions. Newly emerged adults mate in the dark, and the mated females enter the beehive during the evening hours. The GWM can lay more than 500 eggs in its life [8], and the larvae can survive 60 to 90 days depending on nutrition [9] and bee colony strength [7,8,10,11]. The newly hatched larvae can move very fast. Later, they burrow into the comb and search for food via a silken tunnel in the comb [8,12]. This life cycle makes GWM infestations easy to spread and are difficult to control.

Burges [12] reviewed physical, chemical, and biological methods of controlling the GWM. While physical and chemical methods can control GWMs, there are drawbacks. For example, some of these methods can only be used on stored empty combs, and pesticide residues from treatments are often left in the bee products. Therefore, developing a biological method to control the wax moth is the best long-term solution. However, the results of some previous studies showed that the use of *Bacillus thuringiensis* did not appear to be an efficient method [4,12]. Predators (*Solenopsis invicta*, or *Solenopsis germinita*), or parasitoids (*Trichogramma* sp., or *Bracon hebetor*) were also used to control the greater wax moth. However, these natural enemies or parasitoids controlling methods are context-dependent methods for controlling GWMs [4,13,14].

A successful and sustainable biological method still needs to be developed for controlling the GWM. Kwadha, et al. [4] suggested using semiochemicals to control this pest. A number of candidate chemical cues are involved in GWM behaviors, like sex pheromones, aggregation pheromones and so on [8,15,16]. On the other hand, some chemicals play a crucial role in honeybee-GWM interaction, like food odors and bee pheromones [17].

Selection of suitable oviposition sites play crucial roles in Lepidopteran insect reproductive success [18,19]. So far, little is known about how a GWM locates a beehive and chooses a hive for egg laying. Larvae prefer tunneling in the comb with pollen, potentially indicating that food odors are involved in GWM offspring preference [8]. It would be interesting to know if GWM oviposition site selection is mainly based on offspring preference or mother preference/avoidance [18,19].

Chemical communication is of importance to bees in a dark beehive. Bees use alarm pheromones to alert mates or even interspecific bees about the predator risks at foraging sites [20,21]. In the hive, bees bite and sting invaders like moths, releasing alarm pheromones in the process which recruits more workers to join the defense [8,17]. We therefore hypothesized the GWM would use female avoidance strategies to orient to the beehive or select an oviposition site. In the present study, we tested whether bee alarm pheromones are eavesdropped on by the female moth, to orient to a better place or avoid a risky site as an oviposition site. We first analyzed the electroantennogram responses of the GWM (unmated moth, mated moth) to the alarm pheromone of *A. cerana*. Secondly, we tested the performance of the GWM in a Y maze to determine whether they are attracted or repelled by bee alarm pheromones. Thirdly, we analyzed preference/avoidance of the GWM to the bee alarm pheromones for selecting an oviposition site.

## 2. Materials and Methods

### 2.1. Insects

The greater wax moth (*Galleria mellonella* L.) was first collected from the bee colony as large sized larvae (400~560 mg) around the 7–8th instar in age and raised in the lab in an incubator (24 °C, Relative Humidity (RH): 65%). Cocoons were collected carefully and placed in a mating cage after emerging. The eggs were collected after adults mated and laid. The first generation offspring were used as experimental subjects and reared as below: an oviposition card with eggs was introduced into a glass tube with natural bee comb with bee bread as food. They were hatched in an incubator (30 °C, RH: 65%). When they reached the fourth instar stage, all the larvae were distributed into different plastic boxes, each box contained 50 to 60 larvae. These boxes were also placed in an incubator (24 °C, RH: 65%) until they pupated and emerged.

### 2.2. Chemicals

Bee alarm pheromone compounds of *A. cerana* were recently identified in our lab with the main components including isopentyl acetate (IPA, TCI Co. Ltd., Tokyo, Japan), benzyl acetate (BA, TCI Co. Ltd., Tokyo, Japan), and octyl acetate (OA, TCI Co. Ltd., Tokyo, Japan) [20,22]. 2-heptanone (2-HP, J&K Co. Ltd., Beijing, China) was also identified and was shown to be a defensive component secreted from bee mandibular glands when bees bite invaders [23,24]. These chemicals were purchased. All the chemical standards were dissolved into hexane (TCI Co. Ltd., Tokyo, Japan) until 100 ng/μL for subsequent electroantennogram or behavioral tests.

### 2.3. Electroantennogram Responses

To determine if the GWM adults detect the olfactory compounds from bee alarms differently among the virgin adults and mated adults, we tested the antennal responses of different aged female adults (1st, 3rd and 5th day old). Electroantennogram responses were recorded for 10 unmated and mated moths for all four bee alarm components with the concentration of 100 ng/μL, respectively.

To record the antennal response, we first carefully captured the moth, cut off its antennae, and placed the antenna between glass electrodes which were filled with insect Ringer’s solution. The antennae were placed about 1 cm away from the outlet of a Polytetrafluoroethylene (PTEE) tube that provided the pulse-stimulus test odor (5 mL/s, controlled by an automatic timer) and clean (500 mL active charcoal filtered) and wet (distilled water) 15 mL/s continuous air flows. Pulsed test odor air flow was delivered into the continuous air flows for 3 s. A custom stimulus controller, a modified EAG amplifier (Syntech, Hilversum, NetherLand, modified to increase sensitivity) was used to output a signal into a Digital Multi-Meter (HP34405A, Agilent, Renton, DC, USA) and BenchVue software running on a PC (Keysight, Renton, DC, USA).

For each antenna, we measured its response to these four alarm compounds and hexane as a control chemical in a pseudo-random order: (1) Hexane, (2) BA, (3) IPA, (4) OA, (5) 2-HP. To reduce odor interference, a 30 s interval between the chemical stimuli was introduced. Antenna response to each group of GWMs was tested 10 times. The EAG response amplitude for each chemical was recorded from the peak to the trough when the chemical stimulus was applied.

### 2.4. Orientation Behavioral Tests

A glass Y maze (4 cm in diameter, 20 cm length for each arm, the end of two odor arms was connected to a 600 mL glass bottle as an odor carrier) was used to determine whether the GWM would be attracted or repelled by bee alarm components (IPA, BA, OA, 2-HP). A total of 10 μL of each chemical was individually applied to a 1 cm^2^ filter paper, placed in one of the bottles at the end of the arm, while the other bottle at the other end of the arm contained an identical size filter paper, which was treated with the same amount of hexane as a control. A charcoal-filtered, wet airflow was applied constantly (speed: 600 mL/min). To stimulate orientation choice in the Y maze, the whole setup was placed in a dark room with a red light. Directional bias in the Y maze without any test odors were pre-tested, and proved no bias if we used only filtered air flows for both arms in the red light condition.

We determined how a moth would choose between one of the bee alarm components and a control arm in a Y maze. GWMs were tested one by one, each moth was observed for 8 min. If a moth entered an arm of the Y maze and proceeded into the odor area of that arm, this was counted as a choice. In some cases, the moth did not reach the end of the arm and returned to the other arms, so the duration of the GWM stay in each arm was also counted with a timer. We calculated how long a moth might stay in the entrance tube, chemical treatment tube, and the control tube, respectively, and also calculated the percentage to standardize all choices from different moths. Each moth was tested once for 8 min, while the moths that stayed in the entrance tube without moving for more than 4 min were excluded from sequential analyses. After each trial, the Y tube was cleaned with ethanol. To avoid the direction bias, the placement of the treatment tube was changed every trial. Fifteen moths were used in each treatment, three different ages (1st, 3rd and 5th day after emergence) of moths were repeated.

### 2.5. Oviposition Sites Preferences

In natural conditions, the female moth looks for an oviposition site after mating. We carried out the experiments in the lab to see whether the moth would avoid laying eggs in places containing bee alarm pheromones. The newly emerged moths were first separated into different tubes (diameter 3 cm, length 11 cm) for one day to become mature and avoid mating competition. One pair of male and female moths (2 days old) were put in a plastic box (5 × 10 × 5 cm^3^) around 10 pm for mating during the night. The following morning, the male moth was transferred to the other box, therefore only the female moth would remain to make an oviposition choice. Two egg laying cards, identically made from a folded filter paper glued with a nylon net (4 × 4 cm^2^), were attached to the opposite sides of the box; one card was treated with test odor, the other card was treated with hexane as a control. Each card was left for one day, and chemicals were added every day. A total of 10 μL of each chemical was added each time if there were still no eggs laid in the box.

The number of eggs was counted for the first day and the second day, to determine the preference of each odor for the female moth for oviposition. We calculated the percentage of egg laying on the treated filter paper to the sum of both papers. Each test odor was repeated eight times. Female moths that did not lay any eggs during these two days or any eggs laid afterward were excluded from sequential analysis.

### 2.6. Statistics

Electroantennogram response data were analyzed with a univariate ANOVA using days, different bee alarm components, and mating state (unmated vs. mated) as the main fixed factors, respectively. Least significant differences (LSD) were used for post hoc tests to determine if there were differences between unmated and mated GWM groups, different alarm components groups and different day moth groups.

Since the duration of GWM choices in the Y maze experiment varied a lot between each individual, they were calculated and presented as a percentage. The choice to the alarm component arm (one out of three arms, such as entrance arm, control arm and the alarm component arm) was compared with the expected value of 33.33% with Chi-square tests. The oviposition site choice experiments were also calculated as a percentage, and the choice of the alarm component filter paper was compared with the expected value of 50% with Chi-square tests.

## 3. Results

### 3.1. Electroantennogram Responses of Unmated/Mated Female GWM

The electroantennogram responses of the greater wax moth to bee alarm pheromones were reduced in intensity as adults aged. The first day moths showed the highest EAG response (more than 4~6 mV), while the third day moths showed the medium EAG response (range from 2~5 mV), and the lowest EAG performances in the fifth day moths were less than 2 mV to all four components of bee alarm pheromones (BA: F_2,27_ = 24.25, *p* < 0.001, OA: F_2,27_ = 14.186, *p* < 0.001, IPA: F_2,27_ = 17.928, *p* < 0.001, and 2-HP: F_2,27_ = 19.41, *p* < 0.001). Similar results were also noticed in mated moths that also reduced with age (F_6,216_ = 0.694, *p* = 0.655). The mated moth also showed lower EAG responses than unmated moths (F_1,216_ = 18.795, *p* < 0.001). No matter which day, unmated moths always showed higher physiological performance than mated moths in BA (F_1,58_ = 10.28, *p* = 0.002), IPA (F_1,58_ = 8.824, *p* < 0.004) and 2-HP (F_1,58_ = 7.85, *p* = 0.007), except OA (F_1,58_ = 0, *p* < 0.99) (Figure 1).

The electroantennogram responses of the greater wax moth to different odors showed different levels of response, in the order of OA > BA > IPA > 2HP (F_3,216_ = 27.5, *p* < 0.001). Through multiple comparisons among these odors, we found that the moths’ responses to OA were significantly higher than BA (*p* < 0.001), and BA was significantly higher than IPA (*p* = 0.008) and 2HP (*p* = 0.01). There was no statistical difference found between IPA and 2HP (*p* = 0.947) (Figure 1).

### 3.2. Orientation in Y Maze

Our experiment set out to determine whether GWMs would avoid the site which contained bee alarm pheromones. No matter at which concentration of the four compounds, GWM females showed almost no significant preference or avoidance (Appendix A), except in a few cases where GWMs showed avoidance to 100 ng/μL BA in the 3rd day moths (χ^2^ = 20.112, *p* < 0.001) (Figure 2).

### 3.3. Oviposition Site Preference and Avoidance

The number of eggs laid by female moths on each oviposition card during Day 1 and Day 2 were counted, and we calculated the percentage of eggs laid on the alarm pheromone treated cards to the sum of both cards. For each chemical pair, we tested eight mated female moths. These mated female GWMs showed similar trends on Day 1 and Day 2, that they laid more eggs on treatment filter paper with BA (Day 1: 61.38 ± 11.28 (%); Day 2: 52.87 ± 11.75 (%)) and IPA (Day 1: 58.12 ± 8.95 (%); Day 2: 59.52 ± 9.48 (%)), but less were recorded on OA (Day 1: 44.77 ± 10.71 (%); Day 2: 44.56 ± 10.78 (%)) and 2-HP (Day 1: 44.56 ± 10.78 (%); Day 2: 42.24 ± 9.13 (%)) cards compared with the control ones. When compared with the expectation of 50%, mated female moths were slightly attracted to BA and IPA, but were repelled by OA and 2-HP during Day 1 and Day 2 (Figure 3, Appendix A). The oviposition site choices were similar even if we sum the number of eggs laid for the first two days. The moths that would not lay any eggs in these two days were excluded from our analyses.

## 4. Discussion

Inside a bee hive in the dark, bee alarm pheromones play important roles in bee communication, especially orientation and defense [25,26]. In the present study, we found that these chemicals (BA, OA, IPA, and 2-HP) caused high electroantennogram responses in the female GWM. Since the female moth can live only 5 to 10 days, they reduced their physiological responses to these chemicals as they aged (Figure 1). However, GWMs showed no significant avoidance to these four alarm pheromones in orientation behavior (Figure 2) or oviposition site choice (Figure 3).

Age and life cycle stage of moths play crucial roles in their behavioral plasticity and pheromone perception. Neilsen and Brister [6] extensively observed the behavior of GWM adults for around 5 years, and showed GWMs emerged from the beehive but soon ran out of the hive entrance (1st day adults). Later, they flew into trees or other high places during dark for mating (1st to 2rd day adults). This phenomenon indicates that the GWM adults would be more active during the first three days, and later, may reduce their activities as they age. In the present study, we consistently tested the physiological responses and behaviors of GWM adults in 1st, 3rd and 5th day moths respectively. GWM physiological sensitivity decreased as they aged (Figure 1). Additionally, most of the female GWMs laid eggs in the first two days after mating. These findings were consistent with the age of drones decreased responses to queen mandibular pheromones [27]. Seven types of antennal sensillia of newly emerged GWMs were observed via scanning electron microscope [28]. How these sensillia develop and dynamically sense the odors in environments as the insect ages still needs further study.

Semiochemicals are abundant and play crucial roles in animal communications in dark hives [29]. Queen pheromones are used to maintain the social order of the colony [30]. Bee alarm pheromones play differential roles in different contexts, such as risk alerts to prevent hive mates from foraging at flowers with dangers [20], or to recruit bees to the entrance [31]. However, in the hive, the alarm pheromones are also used to defend against parasitoids [17].

The chemicals work as a private signal between bees themselves [14], and can also be eavesdropped on by other animals [16]. For example, *A. cerana* benefit from eavesdropping on alarm pheromones from *A. dorsata* and *A. mellifera* in order to avoid dangers [21]. In addition, parasitoids, *Bracon hebetor*, can locate their hosts via eavesdropping on the sex pheromones of GWM adults [14]. Based on the physiological data, both mated and unmated female GWM adults can sense the four components of bee alarm pheromones (Figure 1). This was similar to the results found in a recent study, that the GWM can detect IPA [17]. We further found that the GWM also reacts to the other components of bee alarm pheromone. Interestingly, GWM females react to octyl acetate (OA) more strongly than IPA and BA, even though IPA and BA were shown to be the main alarm components in *A. mellifera* and *A. cerana*, respectively [22,26].

GWMs also communicate with pheromones, like male produced sex pheromones [32,33,34]. However, some of wax moth pheromones, like nonanal, decanal, and undecanal are also commonly found in the beehive [35]. These chemicals in the beehive may alter GWM neural and behavioral responses as well [36,37]. In the present study, we tried to find out whether GWMs would avoid risky sites for egg laying. For the animal oviposition site choice hypothesis, they either choose it to benefit themselves (the female moths in the present study), or choose it to benefit their offspring (the eggs or larvae for next generation) [18,19]. When bees find invaders like small hive beetles or wax moths, it will trigger their defensive behaviors like imprisonment, biting and stinging the wax moth larvae [4,8,38]. During these aggressive defenses, bees easily release alarm pheromones which contain compounds like IPA, OA, 2-HP and BA [22,25,26]. IPA is the most common alarm component in all *Apis* species [26], but the other components may enhance alarm function by persisting for longer [21].

An alternative hypothesis would be that GWMs could sense the bee alarm pheromones like they could sense all other semiochemicals in the beehive [17], but not take alarm pheromones as risk-related information. GWMs mainly invade the beehive during the night when most of the bees are resting and less defensive [6]. Newly emerged moths would fly out from the beehive to mate, then select a hive to invade around 2 h after dark, when bees became less defensive. Even some moths that were seen to be caught by guard bees could run away and turn a half circle back to the beehive without stopping [6]. Less or even no direct conflict may happen to drive the GWMs to avoid bee alarm pheromones for their oviposition site selection.

Both the fecundity-survival hypothesis and preference-performance hypothesis suggest that female insects may ignore their own risk to select oviposition sites which can offer higher survival rates or nutritional quality for their offspring [18]. GWMs were observed using their antennae and ovipositors to select a suitable place to deposit eggs. Wax moth eggs were found in pollen cells, or frame corners [6].

## 5. Conclusions

In the present study, both unmated and mated female GWMs showed physiological responses to these four bee alarm pheromones. Unmated GWMs showed higher responses than mated GWMs. This was consistent with the previous study that the pre-mating and oviposition period of the moth would behave differently to risk or other selection pressures like temperature [39]. We even predicted GWM would avoid these risk related cues of bee alarm pheromones based on physiological results, however, both orientation and oviposition results showed no significant avoidance or preference. The results indicate that bees seem to lose the arms race since GWMs successfully invade the beehive, reproduce in bee combs, and honeybees are not able to expel them, or even police their laid eggs. Honeybees release alarm pheromones, which the greater wax moth can sense but seem to ignore.

## Figures and Tables

**Figure 1 insects-10-00081-f001:**
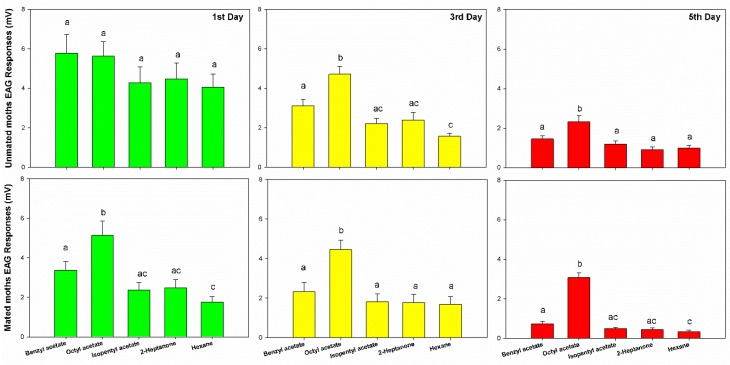
The electroantennogram responses of female moth (unmated moth *n* = 120, mated moth *n* = 120) to bee alarm pheromones (benzyl acetate, *n* = 60, octyl acetate, *n* = 60, isopentyl acetate, *n* = 60, 2-heptanone, *n* = 60) in 1st day (*n* = 80), 3rd day (*n* = 80) and 5th day (*n* = 80) moths, error bars are presented with Mean ± SEM. Different letters indicate statistical significance in the bar graph.

**Figure 2 insects-10-00081-f002:**
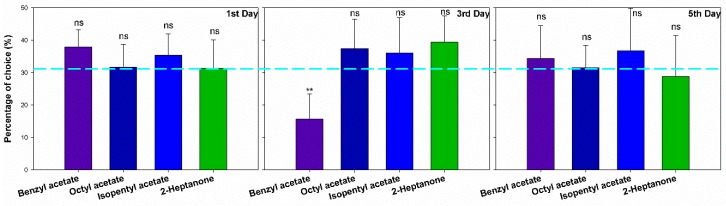
The choice of greater wax moths (1st day *n* = 60, 3rd day *n* = 60 and 5th day *n* = 60) to different bee alarm pheromones in the Y maze. The light-blue reference line showed 33.33%, error bars presented with Mean ± SEM. “ns” means no significant difference, “**” means significant difference (*p* < 0.01).

**Figure 3 insects-10-00081-f003:**
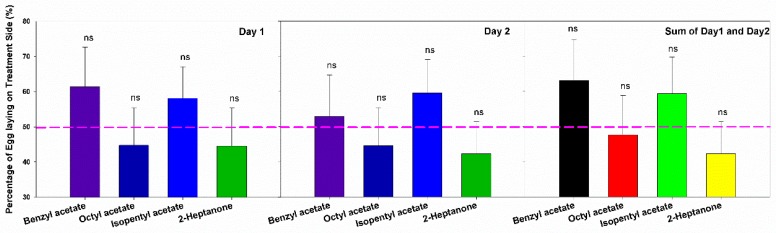
Oviposition site selection between the bee alarm pheromones and Control (benzyl acetate: *n* = 8, octyl acetate: *n* = 8; isopentyl acetate: *n* = 8, and 2-heptanone: *n* = 8). The pink reference line showed 50%, error bars presented with Mean ± SEM. “ns” means no significant difference.

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
