# Peer review of "Losing the Arms Race: Greater Wax Moths Sense but Ignore Bee Alarm Pheromones"

_insects, 2019, doi:10.3390/insects10030081_

Round 1

Reviewer 1 Report

REF: insects-442665

Loosing the Arms Race: Sensed but Ignored by Greater Wax Moth on Bee Alarm Pheromones

Overall, this is a nice study which deserves publication in INSECTS. I found the results interesting and the conclusions are compatible with the evidence and arguments.  However, the MS needs to be checked either by a native speaker or a person with full professional proficiency in English.

Author Response

Response to reviewers:

First of all, we would like to thank both reviewers carefully reading through our manuscript. We carefully addressed to both reviewers’ comments point by point as below. All changes could be easily found in yellow in the manuscript. And the answer to each point was followed with a bold Response.

Response to Reviewer 1:

Losing the Arms Race: Sensed but Ignored by Greater Wax Moth on Bee Alarm Pheromones

Overall, this is a nice study which deserves publication in INSECTS. I found the results interesting and the conclusions are compatible with the evidence and arguments. However, the MS needs to be checked either by a native speaker or a person with full professional proficiency in English.

Response: Thanks for your encouragement and the comments, we again asked our co-author, Ms. Katrina Klett who is a native English speaker to carefully read through the manuscript and polish the English. All the changes were marked in the main text in yellow.

Reviewer 2 Report

The authors have presented a detailed study, investigating the impacts of bee alarm pheromones on greater wax moth. The study is comprehensive and unique, but it lacks some critical explanations to help understand the experiments better.

Most importantly, the manuscript has to be thoroughly edited for English grammar and typos. The title itself has a spelling error (Losing not Loosing). The title also needs to be rephrased. Throughout the manuscript, this issue has resulted in the authors being unable to put their points across. This will also potentially lead to misinterpretation of the manuscript. In addition, there are many related published studies. The authors need to expand their list of citations by including these important, previously published researches in the introduction and discussion sections. Overall, the study adds to the volume of knowledge in understanding responses of wax moths to bee pheromones. However, the following concerns must be addressed.

Abstract: The abstract has to be thoroughly revised. Please do not use abbreviations in the abstract. The abstract, in its present form, has critical grammatical errors.

Introduction: As before, please revise the entire introduction for grammar. Citations needed for L35 (pupae progress through 8 stages). Please spell digits less than ten. L40-L41 also need citations. L53 please use “Kwadha et al. [11]” or “Previous study [11]”. Please rephrase the statement in L60-L61 to end in a period. Please rephrase L47-L51 and L71-72. There is a large volume of previously published research, regarding the different chemicals, which the authors chose for their study. The introduction section is missing sufficient justification for the authors’ choices, based on previously published studies.

Methods: Please rephrase and check for grammar throughout the methods. Methods have not been adequately detailed. L74-L77: Rearing conditions have not been explained in clear details. Please elaborate “big larvae”. Chemicals, equipment, reagents – all have to be clearly mentioned (make, model, details, company name, city/state, country). For electrophysiology, details are not adequate. Please mention clearly, if both males and females were tested (L88). Details of EAG amplifier needed. L96 what is the carry odor? Chemical concentrations needed for chemicals tested. Please use abbreviations after their first mention in the manuscript. What is the Y-maze made of? Did the authors test for directional bias in the Y-maze without any pheromone in the very beginning? L116-L117 please rephrase. For oviposition preference studies, details of egg-laying cards are required. What is the concentration of each chemical that was added in the oviposition cards? Why a mixture of the various chemicals were not tested? Please mention very clearly N, stage of life cycle and sex of the moth tested (male or female) for every experiment. Why was the time duration data in Y-maze choices presented as percentage? Why authors selected expected values of 33% and 50%? Without a clear understanding of the samples tested, the results are difficult to interpret.

Results: Same as before, please rephrase the entire results section and check for grammar. Did the authors perform mean correction of EAG with respect to hexane control? Where is that data? Where is the dose-response curve for EAG? What was the EAG response amplitude for chemicals that were tested? The authors need to present both the degrees of freedom for F in the results. What do the authors mean by “clear airstream” and “chemical airstream”? L160-L166 and L172-L178 ideally should be in methods. What about the electrophysiological and Y-maze results for male wax moths?

Figure legends:All the figure legends need to be rephrased and edited. Please clearly mention N, error bars details (SD or SEM), details of the adult wax moths (age; conditions of rearing; male and female). Please indicate statistical significance in the bar graphs. L242-L243: The authors did not conduct a thorough physiological study. Also, please mention if unmated female/male GWM showed differential responses than mated female/male GWM.

Discussion: Please rephrase and edit the discussion section for typos and grammar. Kwadha master’s thesis has not been added to the reference list. Please maintain similarity when citing previous studies in the main text. Please add more relevant citations to strengthen the discussion. Age, life cycle stage and sex of the moths with respect to antennal structure and pheromone perception have not been discussed in sufficient details.

Author Response

Response to reviewers:

First of all, we would like to thank both reviewers carefully reading through our manuscript. We carefully addressed to both reviewers’ comments point by point as below. All changes could be easily found in yellow in the manuscript. And the answer to each point was followed with a bold Response.

Response to Reviewer 2:

The authors have presented a detailed study, investigating the impacts of bee alarm pheromones on greater wax moth. The study is comprehensive and unique, but it lacks some critical explanations to help understand the experiments better.

Most importantly, the manuscript has to be thoroughly edited for English grammar and typos. The title itself has a spelling error (Losing not Loosing). The title also needs to be rephrased. Throughout the manuscript, this issue has resulted in the authors being unable to put their points across. This will also potentially lead to misinterpretation of the manuscript. In addition, there are many related published studies. The authors need to expand their list of citations by including these important, previously published researches in the introduction and discussion sections. Overall, the study adds to the volume of knowledge in understanding responses of wax moths to bee pheromones. However, the following concerns must be addressed.

Response: Thanks for your kind and careful read of the manuscript. You have raised so many valuable comments to improve this manuscript. Sorry about the typos in the title, the title has been changed as suggested. Our co-author Ms. Katrina Klett has carefully read through the manuscript and revised the English grammar and typos. More detailed information was added in Methods, Results, and Discussion parts to make it clearer. We addressed your comments point by point as below.

Abstract: The abstract has to be thoroughly revised. Please do not use abbreviations in the abstract. The abstract, in its present form, has critical grammatical errors.

Response: The abbreviations in the abstract were changed as suggested, such as Line 13, line16, line17, ling21, line22 and line 24-25. The grammatical errors were corrected.

Introduction: As before, please revise the entire introduction for grammar.

Response: We revised the entire introduction for grammar as suggested.

Citations needed for L35 (pupae progress through 8 stages).

Response: The citation of Yang et al. 2017 was added.

Please spell digits less than ten. L40-L41 also need citations.

Response: Numbers less than ten were spelled as suggested

L53 please use “Kwadha et al. [11]” or “Previous study [11]”.

Response: We rewrote this citation.

Please rephrase the statement in L60-L61 to end in a period.

Response: This statement was rephrased as suggested.

Please rephrase L47-L51 and L71-72. There is a large volume of previously published research, regarding the different chemicals, which the authors chose for their study. The introduction section is missing sufficient justification for the authors’ choices, based on previously published studies.

Response: Thanks for the suggestion, these two parts were rephrased as suggested.

Methods: Please rephrase and check for grammar throughout the methods.

Response: We revised and checked again carefully for grammar throughout the methods part as suggested.

 Methods have not been adequately detailed. L74-L77: Rearing conditions have not been explained in clear details.

Response: One paragraph was added as L79-L85 to explain in clear details how we collected eggs and the rearing process for the first generation of moths for the experiment.

Please elaborate “big larvae”.

Response: big larvae represent the larvae size around 400-560mg, which is close the cocoon stages, now we changed it as: large size larvae (400-560mg) in L79.

Chemicals, equipment, reagents – all have to be clearly mentioned (make, model, details, company name, city/state, country).

Response: chemical standards were labeled with their company name, city, country as L89-L91.

For electrophysiology, details are not adequate.

Response: We added more details about the electrophysiological experiments.

Please mention clearly, if both males and females were tested (L88).

Response: We did the experiment with both males and females, but in the present study we mainly want to focus on female adults, since female adults were the only gender who would invade the beehive and cause damages after they laid eggs.

Details of EAG amplifier needed.

Response: Details of EAG amplifier was added as suggested, (Syntech, NL, modified to increase sensitivity).

L96 what is the carry odor?

Response: The carry odor was active charcoal filtered air mentioned in L103-104, here we changed “carry odor” to “continuous air flows”.

Chemical concentrations needed for chemicals tested.

Response: We used the fixed concentration of 100ng/μL for all these chemicals, mentioned in L94.

Please use abbreviations after their first mention in the manuscript.

Response: We changed chemicals name in abbreviations as suggested.

What is the Y-maze made of?

Response: It was a glass Y maze, now we have added this into the text as suggested.

Did the authors test for directional bias in the Y-maze without any pheromone in the very beginning?

Response: yes, we did this in our pilot experiment, which proved there was no directional bias if we used only filtered air flow for both arms. We added a sentence to make it more clear as: Directional bias in the Y maze without any test odors were pre-tested, and proved to contain no bias if we used only filtered air flows for both arms in red light condition.

L116-L117 please rephrase.

Response: We rewrote this part.

For oviposition preference studies, details of egg-laying cards are required.

Response: we added the details of egg laying cards as suggested as: Two egg laying cards, each card was identically made from a folded filter paper glued with a nylon net (4×4 cm2), were attached to the opposite sides of the box, one card was treated with the test odor, the other card was treated with hexane as a control.

 What is the concentration of each chemical that was added in the oviposition cards?

Response: Ten μl of these chemical was added in each oviposition card, now we added one sentence in the text to make it more clear.

Why a mixture of the various chemicals were not tested?

Response: Thanks for the suggestion, you were right, the responses of moths to the mixture of the various chemicals would be very important for the chemical ecology study, we do have this design, but for mixture of various chemical experiments, we need to take lots of characters into consideration, such as the total amount, the ratio of each chemicals in the mixture and the natural concentration of the mixture. The responses to the mixture will be reported in a future study.

Please mention very clearly N, stage of life cycle and sex of the moth tested (male or female) for every experiment. Why was the time duration data in Y-maze choices presented as percentage?

Response: The time duration of each moth varied a lot, some moths chose the arms very fast, and they could run to the other arms (entrance arm, control arm and treatment arm). That’s why we calculated the duration data in Y maze choices presented as percentage to standardize all choices from different moths. We explained this as well in text L161-L162.

Why authors selected expected values of 33% and 50%? Without a clear understanding of the samples tested, the results are difficult to interpret.

Response: For Y maze experiments, moths could choose among three different arms, the entrance arm, control odor arm and the treatment odor arm, so we selected a value of 33%; while in the oviposition experiments, moths could only choose between the control oviposition card and treatment oviposition card, then we selected an expected value of 50%.

We added a sentence as: The choice of the alarm component arm (one out of three arms, such as the entrance arm, control arm and the alarm component arm) was compared with the expected value of 33.33% with Chi-square tests, to make it more clear.

Results: Same as before, please rephrase the entire results section and check for grammar.

Response: We revised the entire results section and checked the grammar carefully as suggested.

Did the authors perform mean correction of EAG with respect to hexane control? Where is that data?

Response: Now we added the hexane EAG data into the results and the figure 1.

Where is the dose-response curve for EAG?

Response: There is only little activity in the beehive, especially during the night when GWM’s try to enter the bee hive, so we assume the GWM would only be attacked by a few bees or even no attack. So we only used the low concentration of 100 ng/μl

What was the EAG response amplitude for chemicals that were tested?

Response: The EAG response amplitude for each chemical was recorded from the peak to the trough when the chemical stimulus was applied. We added this sentence to the methods part, see L112-113.

The authors need to present both the degrees of freedom for F in the results.

Response: Two degrees of freedom for F in results were added as suggested, see L170-177.

What do the authors mean by “clear airstream” and “chemical airstream”?

Response: we modified this to make it clear.

L160-L166 and L172-L178 ideally should be in methods.

Response: We moved these parts to methods as suggested.

What about the electrophysiological and Y-maze results for male wax moths?

Response: Sorry, we ignored male wax moths, since they won’t mate and return back to a beehive, they almost have no chance to expose themselves to the risk of bee attack. So we mainly focused on female wax moths.

Figure legends: All the figure legends need to be rephrased and edited. Please clearly mention N, error bars details (SD or SEM), details of the adult wax moths (age; conditions of rearing; male and female).

Response: The number of samples (N) were added in each figure legends for each group. The error bars were presented with Mean±SEM, which was also mentioned in each legend.

Please indicate statistical significance in the bar graphs.

Response: In Figure 1, we used different letters to indicate statistical significance in the bar graphs between different groups. In Figure 2 and Figure 3, statistical significance in the bar graphs were added as well, “ns” means no significant difference, “*” means significant difference (P<0.05).

 L242-L243: The authors did not conduct a thorough physiological study. Also, please mention if unmated female/male GWM showed differential responses than mated female/male GWM.

Response: EAG responses between unmated female and mated female were compared, in three different ages (1st instar, 3rd instar and 5th instar). Unmated GWM showed higher physiological responses than mated GWM. We mentioned this in the new text, in the results part (L170-171) and in the discussion part (L262-263).

Discussion: Please rephrase and edit the discussion section for typos and grammar.

Response: We rephrase and revised the discussion section carefully as suggested. Changes can be found in yellow.

Kwadha master’s thesis has not been added to the reference list.

Response: Sorry for these mistakes, we corrected them, please see L234, L241,257.

Please maintain similarity when citing previous studies in the main text.

Response: Changes made as suggested

Please add more relevant citations to strengthen the discussion.

Response: More relevant citations added to strengthen the discussion as suggested.

Age, life cycle stage and sex of the moths with respect to antennal structure and pheromone perception have not been discussed in sufficient details.

Response: A paragraph about how age and life cycle stage of the moths with respect to antennal structure and pheromone perception have been discussed, see L219-229.

Round 2

Reviewer 1 Report

Losing the Arms Race: Sensed but Ignored by Greater Wax Moth on Bee Alarm Pheromones

The MS looks much  improved. However, the language needs to be further polished. I  attempted to help with the Introduction which shows that the rest of the  text also needs to checked and corrected accordingly.

Lines 17-22: Does  the journal require capital letters for chemical compounds such as  Isopentyl acetate, Benzyl acetate and Octyl acetate?

Lines 19-20: I would recommend replacing “a highly electrophysiological response” with “a strong electrophysiological  response”.

Line 24: How about “These results indicate that bees are CURRENTLY losing…”?

Line 28 (Keywords): “Alarm related-risky” – what is this alarm-related risky thing?

Line 34: GWM layS

Line 36: The duration of developmental of

Line 38: up to 12 weeks

Line 39: omit “(personal observation)”.

Line 40: depending on temperature and other conditions. Omit “weather”.

Line 49: in the bee products.

Line 49: Therefore, developing…

Line 50-51: However, the results of some previous studies showed that the use of Bacillus thuringiensis did not appear to be an efficient method [11,12].

Line 52: “were also”. Adverbs usually come after the verb in English!

Line 52: replace “utilized” with “used”.

Line 54: Please, provide a reference/references at the end of this sentence to support your claim “…GWM ()”.

Lines 54-56: Those two sentences bring the same message to the reader. Please, omit one of them to reduce redundancy.

Line 57: Replace “Lots” with “A number”.

Author Response

Response to reviewer 1

Thanks for your kindly and carefully read through the manuscript and raise so valuable comments to improve this manuscript. We addressed your comments point by point as below following with a bold Response. All changes we made could be easily found in the manuscript marked in yellow.

Comments from Reviewer 1:

Losing the Arms Race: Sensed but Ignored by Greater Wax Moth on Bee Alarm Pheromones

The MS looks much improved. However, the language needs to be further polished. I attempted to help with the Introduction which shows that the rest of the text also needs to checked and corrected accordingly.

Lines 17-22: Does the journal require capital letters for chemical compounds such as Isopentyl acetate, Benzyl acetate and Octyl acetate?

Response: we now changed all chemical compounds as isopentyl acetate, benzyl acetate and octyl acetate as suggested throughout the manuscript.

Lines 19-20: I would recommend replacing “a highly electrophysiological response” with “a strong electrophysiological response”.

Response: We replaced “a highly electrophysiological response” with “a strong electrophysiological response” as suggested.

Line 24: How about “These results indicate that bees are CURRENTLY losing…”?

Response: “currently” was added.

Line 28 (Keywords): “Alarm related-risky” – what is this alarm-related risky thing?

Response: We would like to show alarm pheromones could be risky-related chemicals. Now we changed it as “Alarm pheromones” to be clear.

Line 34: GWM layS

Response: “lays” was replaced with “lay”.

Line 36: The duration of developmental of

Response: we now changed it as “The developmental durations for GWM egg, larvae and pupae have been reported to be around…”

Line 38: up to 12 weeks

Response: “up to 12 weeks” was added as suggested.

Line 39: omit “(personal observation)”.

Response: we removed “personal observation” as suggested.

Line 40: depending on temperature and other conditions. Omit “weather”.

Response: we replaced “…depending on temperature and other conditions.” With “weather conditions”

Line 49: in the bee products.

Response: “the” was added.

Line 49: Therefore, developing…

Response: the sentence was changed as “Therefore, developing a biological method to control…”

Line 50-51: However, the results of some previous studies showed that the use of Bacillus thuringiensis did not appear to be an efficient method [11,12].

Response: this sentence was rewrote as suggested.

Line 52: “were also”. Adverbs usually come after the verb in English!

Response: we now reversed the adverb and verb as suggested.

Line 52: replace “utilized” with “used”.

Response: we replaced “utilized” with “used”.

Line 54: Please, provide a reference/references at the end of this sentence to support your claim “…GWM ()”.

Response: reference was added.

Lines 54-56: Those two sentences bring the same message to the reader. Please, omit one of them to reduce redundancy.

Response: “A deep understanding of the biology of these insects themselves is required” was removed as suggested

Line 57: Replace “Lots” with “A number”.

Response: we replaced “Lots” with “A number” as suggested.

Reviewer 2 Report

The authors have performed a significant revision in providing the missing information and the manuscript is considerably improved. Please see my comments about specific sections. I believe this manuscript will contribute to the existing information on greater wax moth. Most importantly, I have concerns about the emphasis/logic behind the study. Wax moth infest abandoned colonies or stored frames. Sometimes they are able to infest live colonies, only when the colonies are weak, diseased or declining. The bees may form a cluster at some part of the frame and the wax moths infest the uncovered parts of the frame. Hence, there is a very high probability that wax moths ignore bee pheromones, as they do not clash with strong colonies at all and stored frames and wax will always have residual bee pheromone smells. I have the following comments which must be addressed:

1. The title still sounds a little off. This is just a suggestion though. Rephrase as: “Losing the Arms Race: Greater Wax Moths Sense but Ignore Bee Alarm Pheromones”. This is because “… on bee alarm pheromones” does not sound right in the title.

2. Abstract: L13 Insert “,” after Galleria mellonella L. L14 please rephrase generating bee products. Instead, use phrases like “The larvae burrow into the wax, damaging the bee comb”. This is because wax moth usually cannot drive bees out. They infest abandoned colonies or stored frames. Sometimes they are able to infest live colonies, only when the colonies are weak. The bees may form a cluster at some part of the frame and the wax moths infest the uncovered parts. L15-L17 still needs to be rephrased. Please do not capitalize the first letters of the alarm pheromone names. The abstract still needs to be corrected for grammar. L19-L26: I think the authors need to rephrase slightly. Use a better phrase than “a highly electrophysiological response”. I believe the authors performed electroantennogram. Electrophysiology sounds misleading.

3. Introduction: L33-L34 Wax moths alone do not cause colony absconding. Please provide references for wax moth larvae burrowing into wax, damaging combs and colony absconding. Without a citation, this information of colony absconding is misleading. Use “stored frames” instead of combs. L36-L37: Please rephrase as: The developmental durations (or use terms like instar phases) for GWM egg, larvae and pupae have been reported to be around ….. I started to edit but realized the introduction still needs a lot of work on the grammar. Citation for L52, mentioning predatory control of GWM. L56 Kwadha et al. Please do not use the entire list of author names. Include references for L57-L58. L63-L64 still ends as a question. Please rephrase to end in a period. L67-L68 need citations. L68-L69 please rephrase.

4. Methods: With the additional of new information, the methods are easier to follow. L83-L84 what is bee pollen comb? Is it bee wax or bee bread or bee pollen? L90 include “and” before ocyl acetate. L96: why floral scents? L98 check for grammar. L144-L145 and L149-L153: please check for grammar. I believe the authors performed electroantennogram and not electrophysiology. Please correct this throughout the manuscript. Why do a univariate ANOVA? Why not do a 2-way repeated measures ANOVA?

4. Results: when the authors indicate instar, do they mean larvae, pupae or adults? If EAG was performed on adults, please do not use the term “instar” and instead use age. Please add data (mean values with SEM) for L188-L197.

5. Discussion: L199-L200 Please rephrase as something like “Inside a bee hive in the dark, bee alarm pheromones play important roles in bee communication, especially orientation and defense (references).” Authors performed electroantennogram. GWM do not show high electrophysiological responses. Instead, they respond to an odor, which was tested via electroantennogram studies. Please rephrase electrophysiological responses throughout the manuscript. L220 add year fro Nielsen and Brister. L219-L229: I am not sure what the authors mean. How is the study, by Neilsen and Brister, related to what the authors want to discuss and have shown in the manuscript? Context is missing. Also, the same for drones’ response to queen mandibular pheromone. Please replace male honey bees with drones. Citation for L230 and L231. L235 rephrase “channel”. L237 “alarm pheromones of different species” of what? L237-L239 check for grammar.

Author Response

Response to reviewer 2

Thanks for your kindly and carefully read through the manuscript and raise so valuable comments to improve this manuscript. We addressed your comments point by point as below following with a bold Response. All changes we made could be easily found in the manuscript marked in yellow.

Comments from Reviewer 2:

The authors have performed a significant revision in providing the missing information and the manuscript is considerably improved. Please see my comments about specific sections. I believe this manuscript will contribute to the existing information on greater wax moth. Most importantly, I have concerns about the emphasis/logic behind the study. Wax moth infest abandoned colonies or stored frames. Sometimes they are able to infest live colonies, only when the colonies are weak, diseased or declining. The bees may form a cluster at some part of the frame and the wax moths infest the uncovered parts of the frame. Hence, there is a very high probability that wax moths ignore bee pheromones, as they do not clash with strong colonies at all and stored frames and wax will always have residual bee pheromone smells. I have the following comments which must be addressed:

The title still sounds a little off. This is just a suggestion though. Rephrase as: “Losing the Arms Race: Greater Wax Moths Sense but Ignore Bee Alarm Pheromones”. This is because “… on bee alarm pheromones” does not sound right in the title.

Response: We now rephrased the title as “Losing the Arms Race: Greater Wax Moth Sense but Ignore Bee Alarm Pheromones” as suggested.

Abstract: L13 Insert “,” after Galleria mellonella L.

Response: “,” was inserted

L14 please rephrase generating bee products. Instead, use phrases like “The larvae burrow into the wax, damaging the bee comb”. This is because wax moth usually cannot drive bees out. They infest abandoned colonies or stored frames. Sometimes they are able to infest live colonies, only when the colonies are weak. The bees may form a cluster at some part of the frame and the wax moths infest the uncovered parts.

Response: we changed the sentence as “The larvae burrow into the wax, damaging the bee comb,…”. But considering wax moths rarely infest live Apis mellifera colony with lots of workers, but mainly infest abandoned colonies or stored frames. But in Asian bee keeping industry, wax moths are seriously damaged A. cerana colony, they could drive the whole colony to abscond. At the same time, they consume the bee bread, leave their feces on their silken tunnel which would degenerating bee products.

L15-L17 still needs to be rephrased. Please do not capitalize the first letters of the alarm pheromone names. The abstract still needs to be corrected for grammar.

Response: we changed all chemical names throughout manuscript. Abstract was corrected for grammar.

L19-L26: I think the authors need to rephrase slightly. Use a better phrase than “a highly electrophysiological response”. I believe the authors performed electroantennogram. Electrophysiology sounds misleading.

Response: we now changed it as “… a strong electroantennogram response” to be more precise.

3. Introduction: L33-L34 Wax moths alone do not cause colony absconding. Please provide references for wax moth larvae burrowing into wax, damaging combs and colony absconding. Without a citation, this information of colony absconding is misleading. Use “stored frames” instead of combs.

Response: Greater wax moths alone could cause Apis cerana colony absconding. We added one citation to support this point. And insert “Apis cerana colony” to be more specific.

L36-L37: Please rephrase as: The developmental durations (or use terms like instar phases) for GWM egg, larvae and pupae have been reported to be around ….. I started to edit but realized the introduction still needs a lot of work on the grammar.

Response: we rephrased as “The developmental durations for GWM egg, larvae and pupae have been reported to be around…” as suggested.

Citation for L52, mentioning predatory control of GWM.

Response: Citations were added, which mentioned red fire ants controlling the GWM.

L56 Kwadha et al. Please do not use the entire list of author names.

Response: We changed this as “Kwadha et al [12] suggested…”

Include references for L57-L58.

Response: References were added.

L63-L64 still ends as a question. Please rephrase to end in a period.

Response: we changed it as “That would be interesting to know GWM oviposition site selection mainly based on offspring preference or mother preference/ avoidance.”

L67-L68 need citations.

Response: citations were added.

L68-L69 please rephrase.

Response: we rephrase this sentence as “We therefore hypothesized the GWM would use female avoidance strategies to orients to the beehive or selects an oviposition site.”

4. Methods: With the additional of new information, the methods are easier to follow.

L83-L84 what is bee pollen comb? Is it bee wax or bee bread or bee pollen?

Response: we cut some old bee comb which contains bee bread as GWM larvae food. here we changed it as “.. with natural bee comb with bee bread as food”

L90 include “and” before ocyl acetate.

Response: “and” inserted

L96: why floral scents?

Response: we didn’t use any floral scents here, so we removed “or floral scents”.

L98 check for grammar.

Response: “differentially” was replaced with “different”.

L144-L145 and L149-L153: please check for grammar. I believe the authors performed electroantennogram and not electrophysiology. Please correct this throughout the manuscript.

Response: We now changed it as “Two egg laying cards, were identically made from a folded filter paper glued with a nylon net, …”

“electroantennogram was replaced with “electrophysiological” throughout the manuscript.

Why do a univariate ANOVA? Why not do a 2-way repeated measures ANOVA?

Response: Two-way repeated measures ANOVA could be used to show the time-related data, like our data we tested on different age of adults (1st day, 3rd day and 5th day). The results could show us a trend either among different age, or different odors, but difficult to show the results like how GWM react to different odors on each day. So we decided to use univariate ANOVA for serval times to show more details in different days.

4. Results: when the authors indicate instar, do they mean larvae, pupae or adults? If EAG was performed on adults, please do not use the term “instar” and instead use age.

Response: All EAG were performed on GWM adults, we corrected” instar” throughout the manuscript.

 Please add data (mean values with SEM) for L188-L197.

Response: The data of Mean±SEM were added into the text.

5. Discussion: L199-L200 Please rephrase as something like “Inside a bee hive in the dark, bee alarm pheromones play important roles in bee communication, especially orientation and defense (references).” Authors performed electroantennogram. GWM do not show high electrophysiological responses. Instead, they respond to an odor, which was tested via electroantennogram studies. Please rephrase electrophysiological responses throughout the manuscript.

Response: we rephrased this sentence as suggested. The electrophysiological responses were corrected throughout the manuscript.

L220 add year fro Nielsen and Brister.

Response: we reformat the this citation as “Nielsen and Brister [7]…”

L219-L229: I am not sure what the authors mean. How is the study, by Neilsen and Brister, related to what the authors want to discuss and have shown in the manuscript? Context is missing. Also, the same for drones’ response to queen mandibular pheromone. Please replace male honey bees with drones.

Response: Here I cited Neilsen and Brister’s works on GWM adults behaviors, which showed that they would be more active during the first few days, later would reduce their activities as aged, that was why we tested the electroantennogram on first 5 days GWM adults.

Now we added one sentence as “This phenomenon indicates that the GWM adults would be more active during the first three days, later, may reduce their activities as aged.” to make it clear.

Citation for L230 and L231.

Response: citations for semiochemicals involved in bee communications were added.

L235 rephrase “channel”.

Response: we changed “channel” as “signal”

L237 “alarm pheromones of different species” of what?

Response: we changed this sentence as “A. cerana benefit from eavesdropping on alarm pheromones from A. dorsata and A. mellifera in order to avoid dangers” to be clear.

 L237-L239 check for grammar.

Response: we changed it as “And parasitoids, Bracon hebetor, can locate their hosts via eavesdropping on the sex pheromones of the GWM adults.”